# Metabolomics and Multi-Omics Integration: A Survey of Computational Methods and Resources

**DOI:** 10.3390/metabo10050202

**Published:** 2020-05-15

**Authors:** Tara Eicher, Garrett Kinnebrew, Andrew Patt, Kyle Spencer, Kevin Ying, Qin Ma, Raghu Machiraju, Ewy A. Mathé

**Affiliations:** 1Biomedical Informatics Department, The Ohio State University College of Medicine, Columbus, OH 43210, USA; eicher.33@osu.edu (T.E.); garrett.kinnebrew@osumc.edu (G.K.); spencer.698@osu.edu (K.S.); Qin.Ma@osumc.edu (Q.M.); machiraju.1@osu.edu (R.M.); 2Computer Science and Engineering Department, The Ohio State University College of Engineering, Columbus, OH 43210, USA; 3Comprehensive Cancer Center, The Ohio State University and James Cancer Hospital, Columbus, OH 43210, USA; Kevin.Ying@osumc.edu; 4Bioinformatics Shared Resource Group, The Ohio State University, Columbus, OH 43210, USA; 5Division of Preclinical Innovation, National Center for Advancing Translational Sciences, NIH, 9800 Medical Center Dr., Rockville, MD, 20892, USA; patt.14@buckeyemail.osu.edu; 6Biomedical Sciences Graduate Program, The Ohio State University, Columbus, OH 43210, USA; 7Nationwide Children’s Research Hospital, Columbus, OH 43210, USA; 8Molecular, Cellular and Developmental Biology Program, The Ohio State University, Columbus, OH 43210, USA; 9Department of Pathology, Wexner Medical Center, The Ohio State University, Columbus, OH 43210, USA; 10Translational Data Analytics Institute, The Ohio State University, Columbus, OH 43210, USA

**Keywords:** multi-omics integration, dimensionality reduction, co-regulation, pathway enrichment, clustering, machine learning, deep learning, network analysis, visualization, biological pathways

## Abstract

As researchers are increasingly able to collect data on a large scale from multiple clinical and omics modalities, multi-omics integration is becoming a critical component of metabolomics research. This introduces a need for increased understanding by the metabolomics researcher of computational and statistical analysis methods relevant to multi-omics studies. In this review, we discuss common types of analyses performed in multi-omics studies and the computational and statistical methods that can be used for each type of analysis. We pinpoint the caveats and considerations for analysis methods, including required parameters, sample size and data distribution requirements, sources of a priori knowledge, and techniques for the evaluation of model accuracy. Finally, for the types of analyses discussed, we provide examples of the applications of corresponding methods to clinical and basic research. We intend that our review may be used as a guide for metabolomics researchers to choose effective techniques for multi-omics analyses relevant to their field of study.

## 1. Introduction

Biomedical researchers are increasingly relying on metabolomics and other omics data types to study and evaluate disease mechanisms and phenotypes. Omics data include, but are not limited to, measurements of the metabolome, proteome, transcriptome, genome, microbiome, and exposome. These measurements include the presence (binary), quantification (abundance), and/or characterization (chemical or biological function) of molecules or entities, such as metabolites, proteins, microbial taxa, genes, or transcripts. For simplicity, we refer to these molecules or entities as “analytes” throughout this work. Multi-omics data may also include descriptors from multiple timepoints in one or more omic modalities, phenotype information such as case/control labels, and relevant clinical variables such as age and sex. Collectively, these data provide holistic insights into disease-driven biological pathway dysregulation, which in turn provides preliminary evidence to drive the identification of new targets or intervention strategies [1]. While the utility of assessing multi-omics data is clear, the integration of metabolomic data with other omic data poses significant computational challenges that range from the need for developing statistical methods that are appropriately adapted to multi-omics integration, to the need for providing comprehensive open-source resources that provide validated relationships between omics types, biological pathways, and diseases. Multi-omics integration typically follows the general workflow depicted in Figure 1.

Ongoing efforts to support the integrative analysis of multi-omics data include the development of statistical methods, computational tools, and pipelines/workflows. Statistical and computational methods comprise novel metrics or novel applications of metrics that describe the relationship between multiple omic data. These include univariate and multivariate analyses, correlation networks, and traditional machine learning and deep learning techniques. A tool is an implementation of a method with proven utility, many of which are designed to be user-friendly software. Tools are often downloadable as an executable file or stored in a public code repository. A series of methods and tools can be combined into workflows to perform an analysis. Supported analyses could span the conversion of raw data (e.g., direct output from instruments or a matrix of un-normalized metabolite and gene levels) to interpretable data that explain a biological system under study. Workflows are particularly useful for conducting repetitive tasks, and typically provide default parameters that are globally applicable, hence making them user-friendly. Examples of open-source and user-friendly workflows include MetaboAnalyst [2], XCMS [3], mixomics [4], miodin [5], and many others reviewed elsewhere [1]. These workflows are particularly useful for end users that may not have a strong data analysis or computational background and are invaluable for outputting reproducible results. Ideally, methods, tools, and workflows provide up-to-date, publicly available datasets that can be used as an input for testing and benchmarking, allowing users to readily evaluate the utility of these workflows for their own purpose.

Many high-level analytical concepts are employed across workflows and data modalities. Understanding the general steps taken by many workflows is crucial to compare the different resources available for performing these tasks. Starting with raw data, prior to analysis, the data quality of each omic data must be carefully assessed to ensure that measurements are reproducible. This step typically requires a comparison of analyte measurements across technical replicates using metrics such as standard deviation or the coefficient of variation. Samples should also be evaluated, making sure that the overall distribution of analyte measurements is consistent across samples. We note that identification of potential outliers (analytes or samples) is critical, as some analytical models for multi-omics analyses (e.g., Principal Components Analysis and Student’s *t*-test) could be strongly affected by the presence of outliers [6,7]. Other preprocessing steps may include normalization to account for differences in experimental effects such as differences in amounts of starting material and batch effects. Data are then typically transformed so that they follow a Gaussian or “Normal” distribution, which is commonly used for statistical analyses. Importantly, as some analyses will not work on missing data, missing values can be imputed. We note that the imputation method used can affect downstream analysis results [8,9], and thus imputation is still an active area of research [10,11]. Finally, noting that the range of values may differ between omic modalities, appropriate scaling (e.g., to a standard deviation of 1, *z*-scores) within and across omic datasets is critical for ensuring that each omic modality contributes to the analyses and that the effect of one omic modality does not dominate all analyses performed [12]. Special scaling considerations should also be taken for time-series data [13].

After preprocessing, omics data can be integrated in multiple ways. One can analyze or model each omic modality separately and then integrate results (a posteriori integration) or one can integrate data for all omic modalities before any statistical or computational modeling (a priori integration) [14]. Depending on which integration approach is utilized, data may be pretreated differently. For example, scaling analyte measurements appropriately within each omic modality is particularly critical when applying a priori integration. Additionally, the sample origin of the multi-omics datasets dictates which integration approach can be used. For example, a priori integration requires the measurements to be collected in the same biospecimens (tissue, blood, etc.) or individuals to allow measurements to be matched to the same sample, while a posteriori integration does not. When the analysis is performed on the same individual but different biospecimens, e.g., genomic data from blood and metabolomic data from urine, we note that it is not possible to evaluate direct relationships between genes and metabolites and how they may relate to phenotype. However, it is feasible to evaluate whether one omic modality (e.g., metabolites) could act as a biomarker for what is occurring at another level (e.g., genome), or one omic modality can be used to corroborate findings (e.g., biological pathways) uncovered in another omic modality [12].

Recognizing that other reviews provide a comprehensive list of available methods, software, and/or workflows [1,15,16,17,18], we instead focus on providing concepts and considerations that are useful when choosing a method or tool appropriate for one’s desired data types and analyses. As such, we discuss existing guidelines in data curation and tool development and describe the building blocks that are used for developing computational tools and workflows, namely unsupervised clustering approaches to assess data quality or separation by sample type, approaches for modeling co-regulations between multiple omic modalities, approaches for identifying multi-omics factors associated with a phenotype (supervised methods), and methods that provide a biological, chemical, and/or disease context to multi-omics data (e.g., pathway analysis). We provide a summary illustration of these approaches in Figure 2.

Further, the review focuses on open-source resources, and we reference example research projects that make use of these resources in the context of metabolomic and other omic data, denoting whether and to what extent the data used is publicly available.

## 2. Open-Source Tool Development and Data Guidelines

The publication of the Findable, Accessible, Interoperable, Reproducible (FAIR) guiding principles for biomedical research data [19] and their adaptation to software development [20] provides clear guidelines to improve data and software infrastructure. Following these guidelines is critical to ensure the reproducibility and reuse of data and software resources. Currently, guidelines for producing tools and workflows exist, yet they are not widely adopted by the bioinformatics community as the tools and workflows are seldom developed by professional software developers [21,22]. Producing detailed documentation, example sets, and maintenance of tools and resources requires substantial resources and efforts, which are difficult to obtain through large biomedical research funding agencies. Open-source code should ideally include clear documentation detailing how to use the tool and providing example inputs to easily test the software. An analysis of publications in Oxford Presses’ Bioinformatics revealed that roughly half of all publications examined had links to a code repository in their abstracts, mostly GitHub [23]. However, it is unknown how many of these repositories include documentation.

We note that the availability of high-quality, well documented, publicly available omics data to use as input for new proposed methods, tools, and workflows is critical to optimize the multi-omics research process. It has been shown that the majority of studies are not reproducible given the information disclosed in manuscripts. This could lead to intangible results. For instance, Begley and Ellis conducted an analysis of 53 high-impact pre-clinical cancer research papers and showed that only six were reproducible given the information provided in the paper, due either to lack of documentation or unpublished data [24]. While standards for reporting data analysis approaches are not well established, standards do exist for disclosing experimental data. Data standards include Minimum Information About a Microarray Experiment (MIAME) [25], the National Center for Biotechnology Information (NCBI)’s Minimum Information About a Next-generation Sequencing Experiment (MINSEQE), Metabolomics Standards Initiative (MSI) [26,27], and Minimum Information About a Proteomics Experiment (MIAPE) [28]. However, compliance is variable. For example, <50% of manuscripts published in journals requiring MIAME compliance actually met compliance [29]. For MSI compliance of metabolomics datasets, 90%–100% of clinical datasets include tissue or biofluid information, yet < 10% of these datasets include information about the ethnicity of the patient, location of collection, and/or volume of sample collection, and reporting of quality control metrics for in vitro datasets are largely missing [30].

Ideally, all multi-omics data from an experiment should be made accessible from the same location; however, the researcher must be aware of the constraints within a repository when submitting data. For instance, some of the most well-known repositories supporting multiple omic modalities are cancer-specific. These repositories include The Cancer Genome Atlas [31], the Cancer Cell Line Encyclopedia (CCLE) [32], the Therapeutically Applicable Research to Generate Effective Treatments Data Matrix [33], the Clinical Proteomic Tumor Analysis Consortium [34], and the NCI-60 Human Tumor Cell Line Screen (NCI-60) [35]. Of these, CCLE and NCI-60 support metabolomics. In scenarios where the data is not appropriate for an existing multi-omic repository, storing the data across multiple single-omic repositories can present an alternative solution. Single-omic repositories include MetaboLights [36] and Metabolomics Workbench [37] for metabolome data, the PRoteomics IDEntifications Database [38] for proteome data, Gene Expression Omnibus [39] and Sequence Read Archive [40] for genomic data, the Encyclopedia of DNA Elements [41] for functional elements in the genome, and the Human Microbiome Project Data Portal [42], MicrobiomeDB [43], and the Human Oral Microbiome Database [44] for microbiome data. When data is spread across multiple repositories, it can be challenging to identify and organize multi-omic datasets that are collected as part of the same study. Efforts that aim to consolidate different data sources as they pertain to publications, such as the Biostudies database [45], are useful for such multi-omic data identification.

## 3. Unsupervised Clustering of Samples to Assess Data Quality or Separation by Sample Type

Unsupervised analyses use algorithms that are agnostic to phenotype to learn the inherent distributions of the underlying data, discover relationships between analytes (regardless of phenotypes), or assess the overall quality of the data. These analyses may be executed on each omic modality separately and the results integrated using a posteriori techniques, or they may be executed on multi-omics data that has been integrated a priori. We describe methods that are commonly used for multi-omics data, especially focusing on the key role of metabolomics data. It should be noted that this list is in no way exhaustive. Examples of applications corresponding to these methods are given in Table 1.

### 3.1. Dimensionality Reduction

The number of dimensions in a dataset generally refers to the number of analytes measured. In multi-omics workflows, the number of dimensions is typically much larger than the number of samples, a phenomenon known as the curse of dimensionality problem. This phenomenon can lead to overfitting in downstream models, where models may not reproduce in other datasets. Reducing the number of dimensions in the dataset can help mitigate this issue. Principal Components Analysis (PCA) is a commonly used method that accepts a matrix of analytes and samples as input, and reduces the dataset to fewer dimensions, or components, that capture the largest variance in the data. The data can be projected on the first two or more components, thereby potentially revealing clusters of samples. By labeling the samples, for example by batch or phenotype, one can identify clusters that may help evaluate data quality. When using PCA, it is important to report the percent variance explained by each component so that the number of components that capture most of the variance can readily be determined. Sample loadings can be incorporated into PCA that reflect the extent of a variable’s contribution to a component. Noting that a component may separate samples by a phenotype or other metric of interest, loadings can be used to identify phenotype-associated analytes in an unsupervised fashion. When the goal is to assess data quality, PCA is performed on individual omic types, and for metabolomics, data collected from different instruments or ionization modes should be evaluated separately. When the goal is to look for separation and loadings, data can be combined, although care should be taken to appropriately scale the data, particularly since the dynamic ranges of metabolomics data can vary greatly and few datapoints could easily dominate the components. Because of potential differences in the variance of analytes from different omic modalities, it is pertinent to factor in contributions of individual modalities to the final loadings. Multi-Omics Factor Analysis (MOFA) [56] and Multiple Co-Inertia Analysis (MCIA) [57] are two techniques for doing so.

Multi-Dimensional Scaling (MDS) is another related, and commonly used unsupervised dimensionality reduction technique. The input for MDS is a distance matrix representing pairwise distances between samples (or analytes) [58]. Examples of commonly used distance metrics include Euclidean distance and 1-correlation for relative abundance data (transcriptomics, metabolomics, etc.), and the Jaccard and the Bray-Curtis for binary data (e.g., microbiome, genomic variants) [59].

T-distributed Stochastic Neighbor Embedding (t-SNE) is another technique used for dimensionality reduction and visualization [60]. Like MDS, t-SNE attempts to produce a lower dimensional embedding of high-dimensional data where distances in the embedding represent similarities between samples. However, rather than using linear correspondence of similarities directly, t-SNE adopts a non-linear adaptive approach based on matching Gaussian probability distributions over similarities. t-SNE can be very helpful for visualizing complex geometry observed in higher-dimensional spaces. However, t-SNE should be used and interpreted with caution, because changes in input hyperparameters (e.g., perplexity) can produce radically different plots, and misleading apparent clusters can result from random data [61].

### 3.2. Clustering

Clustering methods are often used to group samples and/or analytes together by shared characteristics (e.g., abundances, presence/absence). Hierarchical clustering assembles clusters of related samples using either an agglomerative “bottom-up”, or divisive “top-down” methodology [62]. An agglomerative clustering algorithm starts with each observation in the dataset belonging to separate clusters. Each iteration combines clusters based on their similarity, and the algorithm stops when all observations belong to one cluster [62]. In contrast, a divisive algorithm starts with one cluster, which is iteratively divided into many. Hierarchical clusters can be visualized as dendrograms, and users can “cut” the dendrogram to produce a desired and relevant number of clusters.

Other methods require that the user specify the number of expected output clusters prior to running the algorithm. One of these is *k*-means [63], which aims to divide all samples into well-separated clusters, where the number of clusters is specified by the input *k*. A related method, Partitioning Around Medoids (PAM) [64] is similar to *k*-means, but can take dissimilarity matrices as input. Silhouette plots, which measure the ratio of within-cluster similarity to between-cluster similarity for differing values of *k*, can be used to help the user determine the number of clusters that best matches the data [65]. Another metric that can be used to determine *k* is the gap statistic, which computes within-cluster dispersion for each value of *k* and subtracts from it the expected within-cluster dispersion for the same value *k* on a uniform distribution of observations [66]. The optimal value of *k* is the one that maximizes this difference.

Another method, the Self-Organizing Map (SOM) is a single-layer neural network with nodes laid out in a grid [67]. Each node has weights that are trained to emulate the analyte abundance/expression values of a set of observations that are similar to one another. Training is done by a mapping process where each observation is mapped to its best matching unit/node in the grid, and that unit and its neighbors in the grid are updated to reflect the features observed. After training is complete, each unit represents the centroid of its own cluster. SOM requires specification of the grid size (number of units) by the user. The user must also specify the learning rate, or the rate at which node weights update in each iteration of the algorithm, and neighborhood size, or the size of a node’s sphere of influence on its neighbors [67]. Specialized metrics for evaluating the results of SOM include map embedding accuracy [68] and topographic accuracy [69]. We note that Milone et al. developed a specialized tool for integrating the metabolome and transcriptome in plant studies using SOM [70].

While the methods described above result in all points being assigned to a cluster and do not allow overlap between clusters, this is not the case for all clustering methods. Gaussian Mixture Models (GMM) assume that all the data are generated from a mixture of Gaussians, thereby allowing overlap between clusters [71]. GMM require an upper bound on *k*, the number of clusters, and can be evaluated for performance (in terms of information captured) using the Bayesian Information Criterion [72] or the Akaike Information Criterion [73]. Another method, Density-Based Spatial Clustering of Applications with Noise (DBSCAN), is designed to detect clusters of arbitrary shape and automatically excludes some observations from clustering by designating them as noise [74]. DBSCAN does not require specification of the expected number of clusters a priori, although it requires the user to specify the minimum number of observations per cluster and a minimum distance between adjacent observations within a cluster.

### 3.3. Other Machine Learning Methods

Other machine learning methods, such as random forest (RF) and support vector machines (SVM), that are typically run in supervised modes can be used in an unsupervised manner to characterize data structure. RF algorithms combine many different decision trees that are built on a subset of the samples and features (e.g., analytes) [75]. In unsupervised mode, synthetic data are generated, and RF is trained to differentiate between the real data and the synthetic data, where class (e.g., phenotype) labels are randomized. The algorithm then tracks the number of times two samples are placed into the same terminal node by the trees, which is then weighted and used as a distance metric between all pairs of samples. This distance metric can then be used as input to the clustering methods described above. Support Vector Machines (SVM) [76] can also be run in unsupervised mode, reviewed in [77]. The algorithm learns to separate samples with random class assignments and then iteratively changes the class assignments to optimize the classification accuracy. This results in an optimally separated set of clusters, where cluster membership corresponds to the random class assignment learned by the algorithm.

Autoencoders are another type of unsupervised learning method used in omics data exploration; they are used for learning a set of latent variables that can be used to reconstruct data [78]. Autoencoders are a type of neural network in which the number of nodes per layer is highest in the first and last layer and lowest in the middle, where the middle layer is called the encoding and is the representation of latent variables. The encoding can be used for clustering. The reconstruction error can then be used as a measure of performance. While autoencoders can learn the complex latent variables underlying the data, a downside is that autoencoders, like neural networks in general, can be difficult to interpret [79].

Although other types of deep learning approaches to clustering also exist, these have not been used in multi-omics applications, to our knowledge. However, their usage in general bioinformatics research is reviewed in [80].

### 3.4. Time-Series Data

Clustering of time-series omic data is most easily accomplished when aggregating the individual feature profiles into clusters (e.g., clustering individuals which have similar time-profiles of a single metabolite or clustering genes in a single individual based on similar expression changes). When using multi-omic datasets, features from separate modalities can be concatenated. In this case, any of the above methods can be straightforwardly applied to the vector representations of the time-series. Unfortunately, the simplicity of this technique comes at the cost of discarding information which could be gained by considering time as a continuously varying, or even ordered, dimension.

A common way that a continuous notion of time can be directly incorporated into clustering techniques is by fitting a model and then clustering on model parameters. Fitting splines, a type of piecewise polynomial curve, is a popular approach [81,82]. Models based on the assumed statistical properties of the processes which create the time series have also been explored, such as auto-regressive moving average (ARMA) models [83,84] or hidden Markov models (HMM) [85,86]. Machine learning techniques developed to process data collected in one timepoint can similarly process time-series into latent vectors which can then be clustered [55].

Notably, the choice of distance metric used may greatly affect clustering [87], and metrics that consider possible time-lags between series can provide a more biologically relevant notion of profile similarity. For example, Dynamic Time Warping (DTW) [88] aligns timepoints so that the distance between the aligned samples is minimized and Lag Penalized Weighted Correlation (LPWC) [89] includes a penalty based on the length of lag between series.

## 4. Identifying Groups of Multi-Omics Analytes that are Co-Regulated

Assessing relationships between metabolites and other analytes could shed light on the mechanisms that underlie a given phenotype. Researchers may wish to assess relationships within a single omic modality (integrating modalities using further analytical methods) or across multiple omic modalities. These relationships are typically not causative, and statistical associations between analytes do not necessarily capture direct, physical relationships, and often ignore complex relationships such as post-translational modifications and non-linear reaction kinetics. For instance, Camacho et al. observed that correlations between metabolite abundance levels could arise from both metabolites being near chemical equilibrium or from a large concentration response to a common enzyme, whereas negative correlations could result from two metabolites being part of the same moiety-conserved cycle [90]. Nonetheless, it is feasible that associative networks capture associations that are functionally relevant. Examples of applications that assess co-regulations of metabolites are provided in Table 2.

### 4.1. Associative Networks

Associative networks are built to model relationships between analytes. One example of such networks, correlation networks, or “relevance networks” [102], can be used to evaluate relationships between differentially expressed analytes. In these networks, nodes represent analytes, and edges represent significant correlations between analytes (for example, based on a *p*-value and/or effect size cutoff). Such networks can be built to infer groups of analytes within or across omic modalities that could regulate one another.

Alternatively, partial correlations can be used in lieu of correlations, where “partial” refers to the correlation between analytes, while removing the effect of other covariables. For example, partial correlations can be used to identify the major influencers (e.g., enzymes) of metabolites whose levels are highly correlated [103]. They can also be used to sparsify relevance networks by conditioning correlations between each pair of analytes on the values of other analytes in the dataset and retaining only those “relevant” and “independent” correlations that do not depend on the values of other analytes [104].

Identifying clusters of coregulated metabolites can be used as a feature selection step to select analytes of interest. For example, Weighted Gene Co-Expression Network Analysis (WGCNA) is commonly applied to study relationships between clusters of samples in high-dimensional datasets [105]. WGCNA identifies clusters of highly correlated analytes in samples and builds dendrograms of these clusters for the user to investigate further [105]. These clusters can then be evaluated for biological relevance.

### 4.2. Topological Analysis of Networks

Given a large number of analytes in omic datasets, networks produced can be very complex, oftentimes described as ‘hairballs’ [106]. To simplify interpretation, the topology of the network can be evaluated. For example, one can evaluate the significance of nodes by identifying hub nodes or nodes that have many connections and thereby contribute considerably to the topology of the graph. The identification of hubs has been used to study gene essentiality [107] and protein robustness to knockdown [108]. Metrics such as degree of a node (the number of nodes to which the node is connected) or betweenness-centrality of a node (the number of node pairs whose shortest path passes through the node) can be used to find hubs, as discussed in Jalili et al.’s review on the topic [109]. However, the existence of hubs must be interpreted carefully. Hubs may represent analytes that are abundant in a cell or that have interactions with or relationships to many other analytes, but this does not necessarily mean that they are relevant to the experimental context of the study [110].

Global network characteristics are also potentially useful. The small-world structure, i.e., the tendency of nodes to be connected by paths of short lengths, of analyte networks can also be informative for inferring the evolutionary history of a metabolic network [111] and the level of communication between substructures in the network [110]. In addition, topological analysis can produce submodules, which represent tightly connected substructures which are oftentimes biologically relevant [112]. A strict definition of a submodule is a clique, which refers to a group of analytes that all share an edge with all other analytes in that group. This definition has been used in the single-omic context to find groups of correlated microbiome samples in Crohn’s disease [113]. Methods for the detection of submodules that are tightly connected, but that need not be cliques as such, include module graphical Least Absolute Shrinkage and Selection Operator (LASSO), which has been used for gene expression data but can be extended to multi-omics contexts [114] and Louvain community detection [115]. Submodules can also be detected by spectral clustering, which has been applied to expression quantitative trait loci (eQTL) data but could be extended to multi-omics contexts [116], and by clique conductance, defined by the sizes of and connections between cliques in the graph [117]. Multilayer N-Cut (MuNCut) [118] was developed specifically for multi-omics networks, and aims to optimize submodule detection by minimizing the “cut” (i.e., the sum of weights of the edges removed) as compared to the size of the submodules (i.e., the sum of weights of all edges in the submodules) across multiple omic modalities.

## 5. Identifying Multi-Omics Analytes Associated with Phenotype

Identifying analytes associated with a phenotype can be done in a variety of ways, including by finding differentially expressed analytes between sample groups, by exploring relationships between analytes and how these relationships differ between phenotypes, and by modeling the direct relationship between sets of (possibly related) analytes and a phenotype. These methods are described below, and applications corresponding to these methods are given in Table 3.

### 5.1. Identifying Differentially Expressed Analytes (Univariate Statistical Methods)

Hypothesis testing methods determine, for each analyte, whether to accept the null hypothesis that a statistic of the analyte’s distribution (such as mean or variance) is unrelated to the phenotype. These methods can be used for analytes across multiple omic modalities or within a single omic modality. These methods can be either parametric or non-parametric in nature. Parametric tests require the user to input data that fit a distribution, such as a “Normal” or Gaussian distribution, whereas non-parametric tests do not impose requirements on the underlying data distributions. Well-known examples of parametric and non-parametric hypothesis testing include the Student’s *t*-test and the Wilcoxon Rank-Sum test, respectively. While these methods are restricted to comparing two phenotypes, one-way analysis of variance (ANOVA) (parametric) or a Kruskal–Wallis test (non-parametric) can handle multiple phenotypic groups.

Univariate tests result in *p*-values, which quantify the probability of the null hypothesis being correct given the observed data. When evaluating many analytes, *p*-values must be corrected to account for multiple comparisons to reduce the number of false positives. Examples of methods that can be used for multiple comparison correction include the Family-Wise Error Rate (FWER), such as Bonferroni correction, and False Discovery Rate (FDR), which includes the Benjamini–Hochberg procedure [138]. For a review of these methods and the challenges inherent in multiple comparisons in the omics space, we direct the reader to [139]. Particularly relevant for multi-omics analyses, Karathanasis et al. [140] developed a method for combining results of hypothesis testing applied to separate omic modalities (referred to as “partial p-values”) in which an underlying permutation test is used that addresses correlations between the omic modalities. Other methods tailored to multiple comparison corrections in omics data include the tail statistic, which is based on an expected distribution of *p*-values [141,142], and an Empirical Bayes method originally developed for microarrays but applicable to other omic data [143].

It is well known that using a *p*-value cutoff of 0.05 (or other values) is subjective and that the interpretation of *p*-values has been mishandled [144]. It is useful to also consider effect size, such as fold changes, when determining which analytes are most relevant to a phenotype of interest. A common visualization method, the volcano plot, combines both *p*-values and fold change, highlighting analytes that have both high fold change and low *p*-values between two phenotypic groups.

Univariate analysis of omic data in a time series, as opposed to a single timepoint scenario, is complicated by the need to account for multiple timepoints. One approach to dealing with multiple timepoints is to collapse them into a single summary value before testing for differences between groups. Examples of this technique include calculating a per-individual mean, area-under-curve [145], time-at-maximum [145], or slope in linear regression [146]. The summary value can then be tested for significance using any of the above techniques. By treating time as a discrete effect, the existence of differential time profiles can be tested for directly with a two-way extension of ANOVA [147]. Two-way ANOVA directly tests for statistically different distributions by either experimental condition or time point, or, importantly, for a statistically significant interaction between the condition and timepoint. MetATT [148] supports the comparison of time-course profiles while allowing for variability both within and between timepoints, thereby reducing false positives and false negatives. The fitting of more complex curves to time-course data will generally require statistical tests specific to the type of curve fit. For instance, Berk et al. developed a modified F-statistic for significance testing in their smoothing splines mixed effects (SME) model [149].

### 5.2. Multivariate Statistical Methods

Multivariate methods are slightly more complex (and more informative) than univariate methods in that they consider possible dependencies between analytes and the effects of possible confounders. These methods may either be run on each omic modality separately or on integrated omic data. One common class of multivariate method is Partial Least Squares Discriminant Analysis (PLS-DA). PLS-DA can be thought of as a supervised variation of PCA, where instead of projecting the data to dimensions that maximize the overall variance, data are projected to maximize the covariance between the projected data and phenotype. Notably, PLS-DA is prone to overfitting [150], as it will always find a projection that separates phenotypes, even with randomized data [151,152]. Metrics that evaluate overfitting, such as R^2^, Q^2^, number of misclassifications (NMC), and Area Under the Receiver Operating Characteristic (AUROC), must be then be evaluated carefully. These metrics are compared for statistical significance in [153].

Variations of PLS-DA include the Orthogonal Projections to Latent Structures Discriminant Analysis (OPLS-DA) [154], which removes variation in the set of analyte abundances that is unrelated to the phenotype. Another variation is Sparse Partial Least Squares Discriminant Analysis (SPLS-DA) [155], which ensures that the number of analytes contributing to the model is relatively small compared to the total number of analytes in the input. We note that both OPLS-DA and SPLS-DA suffer from the same overfitting drawback as PLS-DA.

Another class of multivariate methods uses linear models [156]. Linear models assume a linear relationship between a continuous response variable (e.g., analyte levels) and one or more independent variables or covariates (e.g., phenotype). These models are learned by minimizing the error between a predicted response variable and the true response variable, and produce weights, also called coefficients, for each independent variable that indicates the influence or significance of independent variables to the response variable. An extension of this type of model is the linear mixed effects model, which assumes that some independent variables contribute random effects to the model, where the coefficient of that independent variable is randomly drawn from a distribution rather than being a fixed coefficient. As with linear models, linear mixed effects models assume that the relationship between a combination of covariates (e.g., age, gender, or batch) and expression or abundance of each analyte is linear. The effect sizes (fold change) between the actual expression levels and those predicted by the linear model can be computed across sample groups to obtain a list of differentially expressed analytes.

Linear models can be extended to include other types of regression models with non-linear functions. In logistic regression models, such as Semi-Parametric Differential Abundance analysis [157], a sigmoidal function, rather than a line, is learned to fit the data. Linear models also may include regularization for enforcing sparsity (i.e., reducing the total number of analytes determined to differentiate between groups) or discouraging overfitting of the model to the data. One such method developed for multi-omics data is collaborative regression [158], in which the model is learned by minimizing error between each omic modality and the phenotype and between linear combinations of separate omic modalities. Additional standard regularization terms include ridge regression [159], which minimizes the sum of squared coefficients, and Least Absolute Shrinkage and Selection Operator (LASSO) [160], which minimizes the sum of absolute valued coefficients; the elastic net combines both regularization terms [161].

We note that, like PLS-DA, linear models and their variations will always learn a model that separates phenotypes, but the model may not necessarily be robust. The coefficient of determination (the proportion of output variance determined by input) or the root-mean square error (the standard deviation of output prediction error) can be used to measure the robustness of the model.

### 5.3. Identifying Analyte Relationships that Differ by Phenotype

Identifying analyte relationships that differ by phenotype can shed light on phenotype-specific mechanisms. Various methods and tools aim to identify phenotype-specific pairs of analytes within one or more omic modalities. For example, DiffCorr calculates correlation coefficients between pairs of analytes within each group and compares correlation coefficients between categorical phenotypes. It does this by transforming correlation differences between phenotypes into *z*-scores to test their statistical significance [162]. The Discordant Method [163] bins analyte pairs with discordant relationships, identified through a mixture model, into categories based on the type of differential relationship (e.g., positively correlated in Group 1 and negatively correlated in Group 2, positively correlated in Group 1 and no correlation in Group 2). Differential Network Enrichment Analysis [164] computes partial correlation networks across multiple phenotypes, and then finds network modules that differentiate between phenotypes. Finally, Sparse Multiple Canonical Correlation Network Analysis (SmCCNet) extends canonical correlation analysis methods by considering phenotypes when evaluating relationships between two omics modalities [165]. Other methods, such as IntLIM, capture phenotype-specific analyte relationships based on linear models that test interactions between a phenotype and an independent variable (e.g., analyte) [166].

### 5.4. Causative (Flux-Balance) Networks

Flux-balance networks are built on experimentally derived associations between analytes for predicting biomass. These networks are particularly useful for predicting the effects of perturbing certain nodes/analytes of the network. Flux-balance analysis relies on the principle that modeling the concentration of biomolecules is mathematically equivalent to modeling flux [167]. These networks thus use a set of equations relating reaction to biomass that can be solved using linear programming to determine which reactions are essential given quantitative (not relative) analyte abundances. While relative abundances are not used to construct these causative networks, they have recently been shown to enhance flux prediction [168]. The current state of flux-balance analysis in the multi-omics space and software implementations for such analyses are reviewed elsewhere [169,170].

### 5.5. Machine Learning Methods for Predicting Phenotype

Machine learning methods can be used to predict phenotype given analytes from a single omic modality or multiple omic modalities that have been integrated a priori. Unlike in multivariate statistics, machine learning methods do not require a priori selection of confounders or multi-analyte dependencies, as they model dependencies directly from the data. Like statistical models, machine learning models make assumptions that differ based on the model type. The machine learning models used in multi-omics integration include both traditional machine learning and deep learning models.

Traditional machine learning methods include Support Vector Machines (SVM) [76] and Random Forests [75]. While the application of these methods can also uncover global data structures in their unsupervised forms, as described in Section 3, we describe here their supervised functionality. SVM assumes that two phenotypes are separated by a hyperplane, which is a linear combination of analyte characteristics (e.g., levels). SVM algorithms learn the hyperplane that optimally separates two phenotypes. SVMs can also be extended to learn non-linear separators using the kernel trick [76], yet we note that these are more difficult to interpret than linear hyperplanes [171]. From SVM models, one can also evaluate the contribution of each analyte to separating the optimal hyperplane, which is calculated as the magnitude of the linear weights of the hyperplane [172,173,174]. Another approach to decipher the analytes most relevant to the model is to consider both the weight and margin between the hyperplane [175].

Another popular machine learning model is the Random Forest (RF) [75], which is based on decision trees. Each tree represents a branched chain of “decisions”, where the decision to branch right or left in the tree is based on one feature (e.g., one analyte) and an optimal cutoff for that feature (e.g., abundance level cutoff). Each decision tree is optimized using metrics such as Gini impurity [176] or information gain [177]. Each RF model constructs many decision trees using subsets of the input samples and subsets of analytes. The predictions of all decision trees are combined into an ensemble to obtain the final output. Like SVM, various metrics exist to determine the influence of each analyte in the RF model. Generally, metrics either evaluate the decrease in model fitness (e.g., Mean Decrease Accuracy [178] or Mean Decrease Gini [179,180]) when features are removed, or compare the influence of analytes on the model using the original values and shuffled values [181].

To evaluate the extent of separation by phenotype, one can use the information retrieval metrics precision, recall, percent accuracy, and F-score, reviewed in [182]. To visualize overall model performance, receiver operating characteristic (ROC) can be plotted [183]. Lastly and like other statistical learning approaches, such as PLS-DA, all models are prone to overfitting the data. To ensure models are not overfitting, samples should be split into training and testing datasets, where the training samples are used to fit the model and the testing set is used to test predictions made by the model. Ideally, a completely independent validation set of samples should be used as an additional model evaluation.

Deep learning models consist of multiple functions of the input data or subsets thereof, that feed into each other in a series of layers, with the outcome (e.g., phenotype) being predicted in the final layer. There are many neural network architectures that can be used for deep learning, including multilayer perceptrons (MLP) [184], convolutional neural networks (CNN) [185], and recurrent neural networks (RNN) [186]. The breadth of each layer, the number of layers, and the function of inputs used at each layer are customizable to a large extent. Neural networks are sometimes referred to as universal function approximators because they can approximate a broad spectrum of underlying models. However, these deep learning models are difficult to interpret and require many samples to accurately train: for example, recent estimations on Monte Carlo simulated data for MLP estimate a requirement of 50 times the number of adjustable parameters in the network (e.g., number of analytes, number of nodes per layer, and number of layers) [187]. Because omic data typically suffer from the curse of dimensionality problem, where the number of samples is far lower than the number of analytes and hence the dimensions, thereby increasing the risk of overfitting, the application of neural networks is a challenge in multi-omics contexts [188]. Nonetheless, neural networks have been successfully applied in some multi-omics studies, as shown in Table 3. Additionally, Yu et al. explored the use of MLP and CNN architectures for classification on 37 transcriptomic and metabolomic The Cancer Genome Atlas (TCGA) datasets, finding that MLP outperformed CNN in this case [189]. Although other deep learning architectures exist in addition to those described here, they have not been applied in the multi-omics context, to our knowledge. However, the use of other deep learning methods in bioinformatics research in general is reviewed in [190].

## 6. Interpreting a List of Phenotype-Related Analytes in the Context of Biology, Diseases, or Chemistry

Identifications of analytes or analyte relationships that reflect a phenotype of interest are typically not useful unless the biological, disease, or other relevant contexts are considered. Common methods to guide the biological interpretation of these data include identifying enriched pathways, and visualizing relationships between analytes. Applications using these methods are outlined in Table 4.

### 6.1. Pathway Enrichment Analysis

Identifying enriched pathways is a common approach to the biological interpretation of differentially expressed analytes, and numerous approaches exist for this type of analysis, each with advantages and disadvantages [201]. Not only do enriched pathways add interpretability to the data, but dysregulations of pathways are also more reproducible across samples than altered levels of individual analytes. [1,202,203]. While published pathways remain the gold standard for context relevance, pathway analysis tools produce relevant results when publications are sparse [204].

Overrepresentation Analyses (ORA) are based on the Fisher’s or Hypergeometric test and are commonly used to identify enriched pathways. Broadly, these methods test the hypothesis that a given pathway is associated more frequently with analytes in the list of interest than would be expected by chance. A major caveat in ORA is the dependence of the result on the background set of analytes (e.g., all analytes measured, or all analytes in a pathway database) used for each pathway [205,206]. In metabolomics, pathway coverage of different metabolite classes is unequal [207]. For example, lipids suffer from lower pathway annotation coverage than other metabolite classes due to the structural complexity of lipid species [207]. Unequal coverage leads to issues in conventional enrichment testing, because the test result is biased towards annotations that are uncommon in the database. When multiple types of analytes are input into ORA analyses, the *p*-values resulting from analysis of each analyte independently can be combined using Fisher’s method [208] or Stouffer’s method [209], which do not penalize an analyte type that has fewer annotations given a particular pathway. This approach is readily available in various software [2,210,211], and an evaluation of both methods is provided in [212]. Other issues of ORA include the erroneous assumption that pathways are independent from one another, and the reliance on an arbitrary statistical cutoff (e.g., *p*-value) to identify enriched pathways.

Another set of pathway-enrichment methods uses Functional Set Enrichment Analysis (FSEA), which is based on the Kolmogorov–Smirnov (KS) test. FSEA methods were developed to address two drawbacks of ORA: statistical cutoffs and sensitivity to background distribution. Rather than using a list of altered analytes as inputs, FSEA takes the entire panel of analytes as input, usually as a ranked list of fold changes, and scores each pathway by an empirically determined weighted Kolmogorov–Smirnov-like statistic. FSEA, however, is more computationally intensive than ORA, and the statistical hypothesis being assessed is less straightforward to interpret. While there are no publicly available implementations of FSEA that simultaneously test multiple omic modalities, the method could theoretically be extended to this application. Examples of FSEA available in single omic modalities include Gene Set Enrichment Analysis (GSEA) [213], Metabolite Set Enrichment Analysis (MSEA) [214], and the Lipid Ontology web-based interface (LION/Web) [215].

Topological scoring techniques that use the structure of networks to infer pathway associations for altered analytes can also be applied. Enriched pathways can be found by mapping differentially expressed analytes onto individual metabolic pathway networks derived from biological pathway databases to determine the global perturbation of the pathway. In these metabolic pathway networks, nodes represent analytes, and edges represent physical or chemical interactions between analytes (e.g., catalyzation, inhibition) as part of a pathway. Each pathway is its own subnetwork. Global perturbation is measured using a combination of standard pathway enrichment and the topology of the network, such as the length of the path between altered analytes and other analytes in the network, betweenness centrality of analytes, and the degree of an analyte. Methods that fall into this category include Signaling Pathway Impact Analysis [216], Pathway Regulation Score [217], Centrality-Based Pathway Enrichment [218], Topological Analysis of Pathway Phenotype Association [219], Topology Gene Set Analysis [220], Clipper [221], and DEGraph [222], which are often used in pathway analysis of gene sets but can also be applied to other analytes. Ihnatova et al. found that the results of these methods differ in sensitivity and specificity when simulated data vary by topological motif size and size of the overexpressed gene set [223]. We note that MetaboAnalyst also incorporates a form of topological analysis called the Pathway Impact Score, which is based on betweenness centrality of differentially expressed analytes in a pathway [2].

Another type of topological scoring method represents pathways as nodes and relationships between pathways as edges in a network. In these networks, differentially expressed analytes are mapped onto pathway nodes. Then, the topology of the network is evaluated to find sets of related enriched pathways. An example of this type of analysis was performed in Zachariou et al. [198]. Lastly, topological scoring can be applied to networks where both analytes and pathways are represented as nodes, and analyte membership in a pathway is represented by edges. Then, analytes of interest are mapped onto this network, and related pathways are found using known membership. This approach is used by FELLA [224], where the authors demonstrate that biologically relevant pathways not found using other pathway analysis approaches can be highlighted using this approach in epithelial cells, ovarian cancer cells, and blood samples of malaria patients [224].

### 6.2. Visualization of Biological Pathways and Networks

Many pathway visualization tools are embedded into biological pathway databases and are designed to visualize one specific pathway at a time. Users can map their analytes of interest, along with analyte abundances or other characteristics, onto these pathways for further investigation. Examples of these types of visualization include OmicsViewer [225], Visualization and Analysis of Networks Containing Experimental Data (VANTED) [226], and PaintOmics [227]. PathMe [228] provides additional flexibility as it incorporates multiple sources related to biological pathways and evaluates crosstalks between these sources. Other tools provide additional flexibility in that they provide a framework for visualizing pathways and/or networks. For example, Cytoscape [229], GraphViz [230], and igraph [231] are very flexible and allow users to upload custom analytes or pathways along with their relationships. PathVisio provides a user-friendly way to draw pathways and to visualize experimental data on these pathways [232].

Other visualization tools represent analyte–analyte interactions outside of the pathway context. For example, OmicsNet [233] combines protein–protein interactions, miRNA–target interactions, transcription factor–target interactions, and enzyme–metabolite interactions from multiple annotation databases to generate a composite network given a list of analytes.

We note that standard formats exist for networks in the multi-omics space. One of these formats is Systems Biology Graphical Notation (SBGN), which includes three languages used for network representation: Activity Flow, Process Description, and Entity Relationship. Each SBGN language includes standardized glyphs and types of information that can be represented in textual annotations [234]. VANTED follows SBGN specifications. Another format is GenMAPP Pathway Markup Language (GPML), which is an Extensible Markup Language (XML)-based format with graphical elements used for storing pathways. GPML is used by some knowledge bases containing graphical information and by PathVisio. Finally, WikiPathways uses the World Wide Web Consortium’s Resource Description Framework (RDF), which facilitates the integration of structured and semi-structured data by creating links between resources [235].

### 6.3. Sources of A Priori Knowledge

For analyses involving pathway enrichment or related analyses, it is important to consider which biological, biochemical, and disease pathway databases should be used. In fact, the coverage of analytes and analyte types differ greatly between databases [236], and the choice of the database used for analysis, such as pathway enrichment analyses, affects results [237]. For this reason, databases that integrate information from multiple sources and for multiple types of analytes have thus been developed. These are particularly useful for performing pathway enrichment using multiple types of analytes as input, as they maximize coverage of pathway annotations [210,211,237,238,239,240]. In addition, the incorporation of biological context (e.g., biospecimen type, species) into pathway analyses has been shown to yield increased specificity of results when compared to functional analysis without incorporated context [241].

#### 6.3.1. Curated and Community Resources

Various efforts are underway to collect, organize, and disseminate information about metabolites and their association with other analytes or types of information (e.g., diseases, biospecimen location, chemical information). Most open-source resources are maintained by curators who manually input and/or review information about analytes and their annotations from the literature. The database curation process relies on domain experts to ensure the accuracy of the information contained in the database. Prominent examples of databases integrating metabolite pathway annotations with other analyte annotations (genes, proteins, microbes, etc.) include the Human Metabolome Database (HMDB) [242], the Kyoto Encyclopedia of Genes and Genomes (KEGG) [243], the BioCyc database series including MetaCyc [244], and Metabolic Pathways Database for Microbial Taxonomic Groups (MACADAM) [245]. Other resources aim to be comprehensive and incorporate information from many database sources, including MetaCyc [244], Pathway Commons [240], PathBank [246], Relational database of Metabolomics Pathways (RaMP) [210], and Pharmacogenomics Knowledgebase (PharmGKB) [247]. We note that the most widely used resources are actively maintained. Therefore, new and updated versions of the database are deployed at varying frequencies, from every several months to yearly. Over time, the accuracy of the databases increases, as new and corroborating knowledge is incorporated.

The integration of multiple sources is challenging, particularly for metabolites, since it is highly dependent on the coverage of information, the confidence in analyte identity, and the accuracy of mapping analyte IDs across databases. Coverage of analytes and other knowledge, such as pathway membership, is important for analyte identification and the retrieval of metadata. However, recent analysis of metabolic networks revealed that mass spectral libraries only covered 40% of these networks [236]. In biological pathway databases, coverage of genes and metabolites also varies, where 12% of metabolites and 67% of coding genes are mappable to pathways [210]. Additionally, the confidence of identity and the level of resolution (e.g., location of double bonds, strain vs. species) available may also affect mapping to pathways [26,27]. To improve the confidence of annotations, users should use IDs, rather than names, to retrieve information on analytes, and should evaluate mapping results for accuracy. In addition, mapping IDs across databases is also a challenge. Different databases use different IDs, with varying levels of information. For example, metabolites represented by International Chemical Identifier Keys (InChIKeys) [248] uniquely map to one metabolite while the commonly used Chemical Entities of Biological Interest (ChEBI) [249] IDs do not [250]. While most databases include links to commonly used ID types, errors could be introduced when mapping IDs from one database to another because of this discrepancy [251]. Standardization of IDs is thus a major challenge that is not completely solved, although it is being addressed by large community-driven initiatives [26,250,252]. Until nomenclatures are fully converged, the community relies on metabolite naming translation services [253]. This issue could be further mitigated by the use of text mining algorithms [254,255].

#### 6.3.2. Computationally Predicted Resources

Natural Language Processing (NLP) methods can be applied to automatically extract knowledge that can be incorporated in multi-omics data analyses and resources. NLP methods mine information from the literature, where tokens (individual or compound words in a document) are analyzed individually and within sentence structures to extract relevant information. One point of consideration in NLP is the dictionary of terms, i.e., the set of possible tokens. The dictionary of terms may be created manually or from existing knowledgebases or literature, if they are available. For instance, the NLP R package Onassis for omics data uses Open Biomedical Ontologies to build its dictionary [256]. In contrast, the Indian Medicinal Plants, Phytochemistry and Therapeutics dictionary was built manually, as there was no electronic resource containing the names of all Indian medicinal plants and their synonyms [257]. In addition to the dictionary of terms, the sources mined must be appropriate. Sources may include abstracts or full-text articles from multiple journals or journal repositories, in addition to other sources, including curated databases, online encyclopedias, patents [258], drug reviews [259], and lab protocols [260]. When considering journals, choosing to use abstracts only may reduce the number of true relationships found [261], but it is often done because of limited accessibility to full text articles. In addition, both the trustworthiness of the source and the context of the article (e.g., toxicology or immunology) should be relevant to the type of information the researcher wishes to extract. Finally, regular expressions must be correctly formatted to define extraction rules for text. Regular expressions are complex patterns of text that can be matched in a document, and they are used to define rules for extracting text to build the knowledgebase. These should be specialized for the task at hand. For instance, Ben Abdessalem Karaa et al. created separate regular expressions to extract causal, preventative, and associative relationships between types of food, genes, and diseases [262], Nikfarjam et al. used a list of key phrases to describe patient responses to drugs in social health networks [263], and Fan and Zhang created several regular expressions to extract patient dietary supplement use from clinical notes [264].

We note that using NLP in the context of biomedical research offers unique challenges that must be considered, which are reviewed in [265]. One of these is resolving words that co-reference the same analyte; Cohen et al. have worked toward solutions specific to biomedical journals [266]. Other work focuses on the task of associating genes with diseases [267] or finding associations between metabolites, proteins, genes, and diseases [258]. Another challenge is document triage, such as finding documents relevant to a context or field of study [268,269,270]. For a thorough review of NLP techniques, as applied to the biomedical literature, we invite the reader to reference [271].

Several NLP-based resources relevant to multi-omics data have been developed. The NJS16 database is a literature-derived database which contains information on the import, export, and macromolecular degradation of metabolites by 570 microbial (bacterial and archaeal) species in three colonic and liver cells [272]. MACADAM, which also contains functional links between microbial species and metabolites, incorporates information from the International Journal of Systematic and Evolutionary Microbiology and Functional Annotation of Prokaryotic Taxa databases, which are derived from the literature [245]. The Drug-Gene Interaction Database (DGIdb) is an NLP-curated database that stores information about mutated genes that could be useful to identify targets for drug development [273]. Another NLP-based resource, which is specifically focused on liver tissue, is LiverWiki [274], a wiki-based knowledgebase containing liver-related genes, metabolites, proteins, protein interactions, pathways, post-translational modifications, and diseases.

Other computational applications aim to predict analyte ontologies, synonym resolution, prediction of molecule interactions/effects, and pathway prediction. For example, ClassyFire [275] provides a taxonomy where compounds are automatically classified into appropriate taxa using a rule-based classification, based on the Simplified Molecular-Input Line-Entry System Arbitrary Target Specification (SMARTS) string and the Markush format. To resolve the many synonyms that can describe one metabolite and mitigate the duplication of analytes used for downstream statistical analyses, PubChem [276] uses an automated standardization technique. This technique works by computing the similarity between two compounds using multiple chemical properties (e.g., atom valence, functional group, and stereochemistry) and merging metabolites with significantly similar properties.

The prediction of a molecule’s interactions and/or effects (e.g., toxicity) can also be automated by using chemical or molecular similarity between the compound in question and another, more well-characterized compound, as is done by Super Natural II [277]. Lastly, to fill the knowledge gap of unknown biological and chemical pathways in all organisms, pathways for unexplored organisms can be predicted using PathoLogic [278]. Specifically, pathways are predicted by first inferring the reactions present based on the identification of enzymes in the organism and then by associating key reactions with pathways.

In some cases, computational prediction and text mining have been used to enrich experiments without inclusion in knowledgebases. In one study focused on finding associations across omic modalities, Fadason et al. found interactions between metabolite-associated single nucleotide polymorphisms (SNPs), metabolites, and chromatin loops (i.e., physical contact between enhancers and promoters) in human blood by combining literature text mining, known drug interactions, Hi-C chromatin interactions, eQTL, and gene ontologies [279]. Additionally, computational predictions can be used to infer associations between analytes. For example, a study by Le et al., used an encoder–decoder neural network to learn novel functional associations between the metabolome and the microbiome in a cohort of paired Inflammatory Bowel Disease (IBD) patients. Using the weights learned using the encoder-decoder neural network to indicate the strength of the relationship between analytes, they uncovered relationships between known IBD biomarkers, such as between *Ruminococcus* and ropane alkaloids and steroidal saponins, and between *Fusobacterium* and bile acids, alcohols, and derivatives [280]. In addition, Morton et al. developed microbe–metabolite vectors (mmvec), a variation of the NLP method word2vec [281], to embed co-occurrence patterns between microbes and metabolites and then infer interactions using the embeddings [282].

We note that as with any computational prediction approaches, there is a margin of error. In this case, this error is difficult to determine because “we don’t know what we don’t know”. Therefore, it is advisable to understand whether resources used are computationally driven, rather than curated through existing and validated experiments. Community-driven resources also bring into question confidence in the user’s entry, as the user’s expertise is often unknown. We note that many resources lack confidence metrics. Lastly, the context of the experiment (e.g., biospecimen location, disease type, etc.) can be utilized to help prioritize analysis results. One example in pathway enrichment analysis is a tool that uses the literature evidence to prioritize the enriched pathways that are returned [204]. Specifically, the algorithm prioritizes pathways that are supported by multiple articles that are related to the same experimental context in the study. When little or no relevant information from the literature exists, then the statistical significance returned from the pathway enrichment analysis method is given more weight in comparison to the literature evidence.

#### 6.3.3. Metrics Used to Define Confidence in Annotations

Confidence in the correctness of an annotation in a knowledgebase can depend on whether there are unknown enzymes or reactions in a pathway or whether a curated annotation has been verified by multiple experts. MACADAM [245] seeks to address the problem of unknown enzymes and reactions using its Pathway Score, based on the percentage of reactions in a pathway that are annotated with an enzyme, and the Pathway Frequency Score, the ratio of annotated enzymes to total reactions. Several other tools include metadata describing the verification of annotations contained therein. In HumanCyc [283], PathoLogic is used to predict metabolic pathways, and pathways are associated with tiers indicating whether they have also undergone manual review. ChEBI has a similar system based on starring: one star means the metabolite entry was automatically curated from a data source, two stars means the metabolite entry was manually processed by a depositor, and three stars means the metabolite entry was manually curated by the ChEBI organization [284]. Finally, WikiPathways [238] has quality tags that can be used to indicate confidence (e.g., *ProposedDeletion*, *WormBase_Approved*, *Reactome_Approved*, and *Hypothetical)*.

Of note, many databases do not include confidence metrics. In this case, the user could map analytes or annotations back to databases which do contain confidence metadata. This can be done using identifiers for analytes or pathways. Alternatively, users can examine the supporting literature for analytes of interest, which can be a tedious process, particularly when many analytes are being considered. Finally, users could assume the same level of confidence for all analytes and annotations in the databases.

## 7. Discussion

Given the complexity and heterogeneity of multi-omics data and experiments, data analysis and interpretation are collaborative efforts, involving biostatisticians, bioinformaticians, molecular biologists, and domain experts (e.g., clinicians, immunologists). Further, given the large number of available methods, tools, and workflows, it is sometimes difficult to select which approach to consider. We recognize that no single method or approach is comprehensive, but rather, approaches and methods are complementary. Applying multiple methods to the same datasets is thus advisable and may corroborate findings or identify novel analyte patterns or relationships.

Aspects that should be considered when selecting methods are: 1) the requirements of input data, including data distributions and types; 2) the biological question being addressed, noting that each tool typically aims to answer a specific question; 3) the availability and metrics of confidence of external resources; 4) ease of use (some methods are hard to implement without computational or statistical expertise); and 5) reproducibility of the results (some methods are stochastic and yield different results when run on the same dataset). Once a method is selected, it is also important to consider which parameters can be modified. To help with parameter selection, most tools provide default parameters to guide the users, although we note that these parameters may not be optimal for all cases. A balance must then be struck between the ease of use of approaches, and the selection of appropriate parameters given the input datasets.

Currently, computational solutions are still lagging behind the rapid influx of molecular data being generated [285]. As computational methods, tools, and workflows emerge, it is important to compare their utility on benchmarking datasets. At present, well curated, publicly available benchmarking datasets are uncommon. Further, emerging computational approaches may not test their performance or complementarity with other tools on the same datasets, making the comparison of multi-omics approaches challenging. Efforts to create readily available data, including different formats of the same data (e.g., before and after data preprocessing) for direct input into new developments would facilitate comparison and an understanding of which approaches to use for which contexts. We also note that developers of methods should disclose the data and code used, along with their publications, to mitigate current deficiencies in reproducibility [286].

One area that we do not delve into deeply in this review is uncertainty in the identification of analytes, which is prevalent for metabolites, but is also relevant for genes and microbes, which rely on accurate sequence reads and alignments. It is feasible for algorithms to take identification uncertainty into account. In computer science, systems with uncertainty in the input, output, or parameters are often called “fuzzy systems”; machine learning methods for fuzzy systems are reviewed in [287]. The representation of uncertainty in visualization tools has also been explored and is reviewed in [288].

The granularity of information that is being received for metabolomics analysis is increasing, as annotations for analytes, experimentally validated or in silico, is ever-growing. This increasing granularity in turn enables the development of more context-specific analyses. For example, multi-omics data-analysis of colon samples can be restricted to analytes that are known to be colonic, thereby removing potential artifacts and false positives in the data. Similarly, metabolic network models can be built per organism to better capture the underlying biology.

## 8. Conclusions

The development of computational approaches that support multi-omics data analysis is still an active area of research. Given the large number of available approaches to such data analyses, the identification of appropriate method(s) for a researcher’s needs is challenging. In this review, we describe concepts and considerations to be made when performing multi-omics analyses, particularly from the viewpoint of which methods, tools, workflows, and resources are available. We discuss the statistical and computational approaches to the tasks of unsupervised clustering, identification of co-regulated groups of analytes, the identification of associations between analytes or groups of analytes and phenotype, and biological interpretation of the relationships between analytes and phenotype, highlighting methods with examples from real-world multi-omics applications in various domains. In addition, we describe a priori sources of knowledge that can be used in biological interpretation analysis as well as points of consideration regarding these sources. Globally, we anticipate a growth in multi-omics data analysis approaches to meet the demands of biomedical research. Such analyses present a unique opportunity for collaborative work amongst different fields, providing multiple viewpoints and knowledge on the same biological system.

## Figures and Tables

**Figure 1 metabolites-10-00202-f001:**
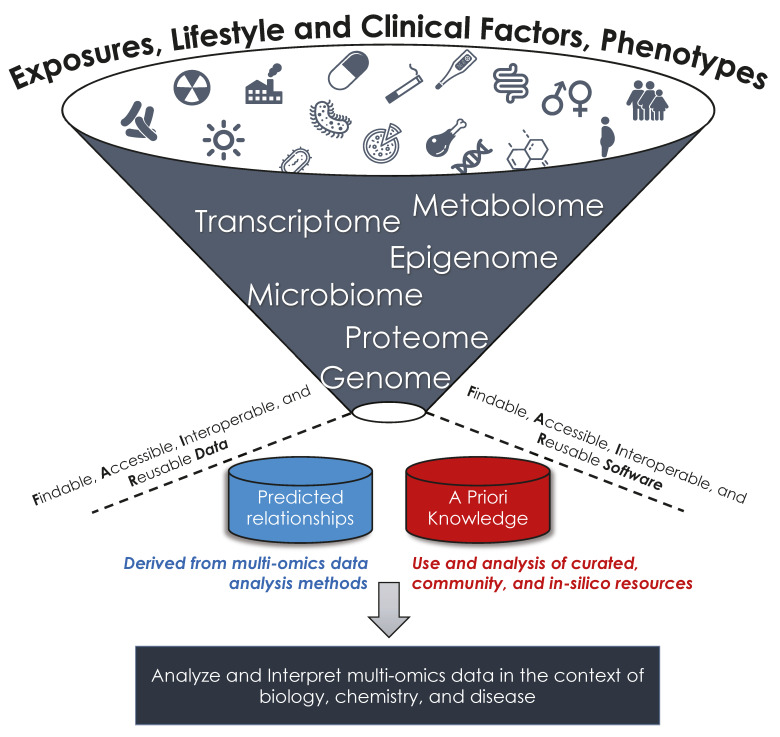
The metabolome in the context of other omics data types and broad approaches for their integration.

**Figure 2 metabolites-10-00202-f002:**
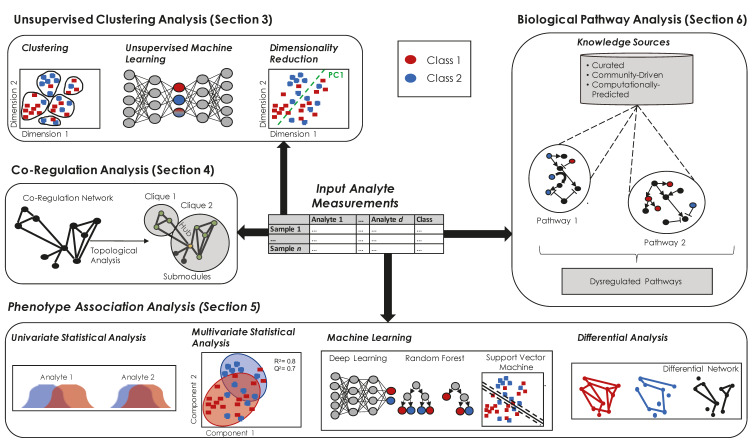
Analysis techniques on a dataset with *n* samples and *d* analytes in two classes. Blue represents one class of samples, and red represents another class of samples. Class typically corresponds to phenotype but could also correspond to batch or another variable of interest. Analyses include unsupervised clustering approaches (Section 3), modeling co-regulation (Section 4), approaches for identifying analytes associated with class (Section 5), and pathway analysis (Section 6).

**Table 1 metabolites-10-00202-t001:** Examples of multi-omics applications using unsupervised analysis.

Type of Method		Functionality	Reference
Dimensionality Reduction	t-Distributed Stochastic Neighbor Embedding (t-SNE)	Visualize gut microbial communities and serum metabolites by diet and supplements.	[46]
		Visualize prefrontal cortex metabolites and lipids by human population group.	[47] †
Clustering	Hierarchical Clustering	Identify multi-omic molecular subtypes in hepatocellular carcinoma.	[48] ‡
		Identify multi-omic clusters in breast tumor tissue associated with prognosis.	[49] †‡
	*k*-means	Identify lipid–protein–metabolite clusters associated with diabetes and periodontal disease.	[50]
	Partitioning Around Medoids (PAM)	Identify microbial–metabolite clusters associated with diarrhea.	[51] *†‡
	Gaussian Mixture Modeling (GMM)	Identify clinical depression score clusters associated with blood metabolomic and genomic data in blood to predict drug response.	[52] ‡
	Density-Based Spatial Clustering of Applications with Noise (DBSCAN)	Evaluate the impact of bacterial metabolism on mucosal immunity.	[53]
Other Machine Learning Methods	Random Forest	Identify clusters of histological stromal features associated with prognosis and metabolites in cancer-associated fibroblasts.	[54] ‡
	Autoencoder	Cluster plasma protein and metabolite levels to identify temporal trends in murine cardiac remodeling.	[55]

* Raw data are available in the supplementary of the referenced manuscript, or a public repository. † Preprocessed data are available in the supplementary of the referenced manuscript, or a public repository. ‡ Descriptive statistics are available in a table or supplementary materials of referenced manuscript. Unmarked data are available upon request from the authors or from a consortium.

**Table 2 metabolites-10-00202-t002:** Examples of multi-omics applications using co-regulation analysis.

Type of Method		Functionality	Reference
Associative Networks	Correlation Networks	Find metabolite–metabolite associations specific to or shared across blood, urine, and saliva.	[91] †
		Find modules of blood metabolites and genes associated with body weight change.	[92] ‡
		Find associations between serum, blood, and gut antibodies, metabolites, and microbiome and patient disease activity reports in inflammatory bowel disease.	[93] *†‡
		Find associations between metabolites, transcripts, cytokines, and cell frequencies in plasma and whole blood associated with adaptive immune response to *Herpes zoster* vaccine.	[94] †‡
	Partial Correlation Networks	Visualize associations between sleep survey responses and levels of serum cytokines, metabolites, lipids, proteins, and genes.	[95] *‡
		Visualize associations between metabolites and lipids associated with metabolic disease treatment in rat liver tissue and clinical chemistry measurements from serum.	[96] †
	Weighted Gene Co-Expression Network Analysis (WGCNA)	Characterize complex transcriptomic and metabolic traits in major depressive disorder.	[97] ‡
		Identify co-regulated modules of blood metabolites and transcripts in children with asthma.	[98] ‡
		Identify co-regulated modules of metabolites and transcripts in glioblastoma multiforme.	[99]
Topological Analysis of Networks	Subnetworks	Identify subnetworks of correlated proteins and metabolites in adrenocorticotropic hormone-secreting pituitary adenomas.	[100]
		Identify subnetworks of correlated genetic, proteomic, metabolomic, clinical, and microbiome data from multiple biofluids in cardiometabolic disease.	[101] ‡

* Raw data are available in the supplementary of the referenced manuscript, or a public repository. † Preprocessed data are available in the supplementary of the referenced manuscript, or a public repository. ‡ Descriptive statistics are available in a table or supplementary materials of referenced manuscript. Unmarked data are available upon request from the authors or from a consortium.

**Table 3 metabolites-10-00202-t003:** Examples of multi-omics applications that identify analytes associated with phenotype.

Type of Method		Functionality	Reference
Univariate Statistical Methods	Student’s *t*-test and effect size	Identify metabolites, miRNAs, mRNAs, and lncRNAs altered by exposure to benzo[a]pyrene to identify mechanisms of toxicity.	[119]
Multivariate Statistical Methods	Partial Least Squares Discriminant Analysis (PLS-DA) (and variants)	Identify breast tumor tissue metabolites that differentiate MRI features.	[120] ‡
		Identify metabolites that differentiate normal and tumor tissue in the prostate.	[121] ‡
		Identify differences between fibromyalgia and control groups in gut microbes, serum metabolites, miRNA, and cytokine levels.	[122] *‡
		Discover temporal changes in plasma lipid and metabolite patterns from normal and hyperlipidemic patients.	[123] †
	Linear Models (and variants)	Identify metabolites from bronchial alveolar lavage associated with continuous CT scan features in cystic fibrosis.	[124] ‡
		Identify serum metabolites associated with visceral adipose tissue features from MRI and tomography.	[125] ‡
		Identify plasma metabolites and proteins associated with prognosis in septic shock patients.	[126] ‡
		Find associations between blood DNA methylation and metabolite levels in smokers.	[127] ‡
Identifying Analyte Relationships that Differ by Phenotype	DiffCorr	Identify differences in metabolite-metabolite correlations between traumatic brain injury and control groups.	[128]
	IntLIM	Identify synovial fluid metabolites and blood and bone marrow transcripts that differentiate between osteoarthritis and rheumatoid arthritis.	[129] *
Machine Learning Methods for Predicting Phenotype	Random Forest	Identify serum metabolites, proteins, and peptides differentiating between metabolic syndrome and control groups.	[130]
		Identify metabolites and other analytes predictive of weight gain and loss.	[131] *‡
		Identify metabolites, transcripts, and proteins predictive of potato quality traits.	[132] †
		Identify metabolites and transcripts predictive of heat stress in the liver.	[133] †
	Support Vector Machine (SVM)	Predict metabolite levels using genes and metabolites in breast and hepatocellular carcinoma.	[134]
	Multilayer Perceptron (MLP)	Predict early and late stage bladder cancer using urinary metabolites and genes.	[135]
		Predict early renal injury using serum metabolites and lipids.	[136] †‡
	Convolutional Neural Network (CNN)	Predict early renal injury using serum metabolites and lipids.	[136] †‡
	Recurrent Neural Network (RNN)	Integrate transcript and metabolite levels to predict cellular state in *Escherichia coli.*	[137] *†

* Raw data are available in the supplementary of the referenced manuscript, or a public repository. † Preprocessed data are available in the supplementary of the referenced manuscript, or a public repository. ‡ Descriptive statistics are available in a table or supplementary materials of referenced manuscript. Unmarked data are available upon request from the authors or from a consortium.

**Table 4 metabolites-10-00202-t004:** Multi-omics applications using biological or visual interpretation methods.

Type of Method		Functionality	Reference
Pathway enrichment methods	Overrepresentation Analysis (ORA)	Identify dysregulated pathways in prostate tumor tissue using metabolite and transcript data.	[191]
		Identify dysregulated pathways in the murine hippocampus and left ventricle during proton irradiation using metabolite and DNA methylation data.	[192]
		Identify dysregulated pathways in cationic liposome treatment of human hepatocyte cells using metabolomic and proteomic data.	[193]
		Identify dysregulated pathways in kidney disease in the rat serum metabolome and proteome.	[194] ‡
		Identify dysregulated gut microbial pathways in gastrectomy patients.	[195] *‡
		Identify dysregulated gut microbial pathways in sports classification groups of Irish athletes.	[196] *‡
		Identify dysregulated gut microbial pathways as a result of whey protein supplementation.	[197] *‡
	Topological Scoring	Identify functional connections between dysregulated pathways in Alzheimer’s using genes, metabolites, miRNA, and proteins from multiple sources.	[198]
Visualization of biological pathways and networks		Visualize metabolic networks in drug-susceptible and drug-resistant strains of *Acinetobacter baumannii.*	[199]
		Visualize interactions between metabolites and genes in non-small cell lung cancer.	[200]

* Raw data are available in the supplementary of the referenced manuscript, or a public repository. † Preprocessed data are available in the supplementary of the referenced manuscript, or a public repository. ‡ Descriptive statistics are available in a table or supplementary materials of referenced manuscript. Unmarked data are available upon request from the authors or from a consortium.

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
