# Peer review of "Metabolomics and Multi-Omics Integration: A Survey of Computational Methods and Resources"

_metabolites, 2020, doi:10.3390/metabo10050202_

Round 1
Reviewer 1 Report
Summary:
This is an outstanding, balanced-depth, and well-written review of the topic of statistical analysis and processing
of multi-omics data. Instead of common reference paraphrasing in review papers, authors have put great effort
to provide a good level of detail. I only have a few minor comments and would like to see this published online
very soon.
Positives:
- Brining focus on reviewing concepts and ideas
- Providing a comprehensive, informative and balanced level of details
- Including useful hints and advice (e.g., multiple methods to the same datasets
Negatives:
- Lacking summary visuals
- Lacking discussion on sources for multi-omics datasets and discussion of relevant examples
Minor comments:
- Overall, Figure 1 is nice but needs some enhancements. The tags (metabolome, etc.) in the middle of the cone
are not clear. Show an example of predicted or curated resources
- "Analytes" term is used frequently in the text. Please provide a definition for it and some
examples of what it might refer to when first mentioned. Example, a variable that might
represent metabolite or mRNAs or genes.
- Authors are advised to include a brief discussion on existing online resources for multi-omics
data. We know that most common resources are for metabolomics, transcriptomics, and other omics
separately. However, some resources do exist with multi-omics data. Can be they used for
golden standard benchmarking/evaluation?!
- Give some biological example and datasets
- "Not only do enriched pathways add interpretability to the data,
536 but dysregulations of pathways are also more reproducible across samples than altered levels of
537 individual analytes. [1,189]."
We have published in our team the following study that might be relevant to cite in this context:
Ewald, Jessica D., et al. "EcoToxModules: Custom gene sets to organize and analyze toxicogenomics data
from ecological species." Environmental Science & Technology 54.7 (2020): 4376-4387.
Please note that this totally up to the authors as they already cited a large set of literature.
- I would have wished for a section on "Biomarker Discovery" techniques (see L1000, S1500 & T1000)
- "Aspects that should be considered when selecting
797 methods are.." --> Maybe, you add reproducibility as one of the guidelines.
Reviewer 2 Report
The authors presented an interesting review dealing with the problems and the need arising in high throughput -omics studies.
I appreciated the depht of investigation of the problems and the high number of references presented.
The only criticism that I can move is about Figure 1: is not clear and som parts are overlappinf, mostly in the higher part. Please change it in order to obtain a clearer picture.
Reviewer 3 Report
Eicher et al presented a review on approaches for data integration with special focus on metabolomics. The integration of multiple omics data is clearly the next step to improve the contribution of omics science to our understanding of biology. Metabolomics offers deep insights into phenotype making it central to the implementation of personalized medicine strategies in health and disease. To take full advantage of such data, they must be integrated with other omics within a consistent framework to enable improved diagnostics, interventions etc. So I am generally very supportive of the review papers on multi-omics as this is a very active area of research and needs to be summarized for practitioners who are willing to take advantage of the great developments in the field. I do appreciate the effect which the authors have put into writing the manuscript, however, I do think that this manuscript is not ready for publication. Unforutenaly, except for a handful of points, most of the manuscript has been written very broadly with no specific focus on multi-omics but rather general methods some of which are used by metabolomics community. Most of the description of the methods are very shallow without a particular description of the assumptions, pros and cons etc. I recommend rejection of the manuscript at this stage but I do encourage the authors to rewrite the manuscript with a much narrower focus on state of the art methods in multi-omics. Please double-check that the description of the methods is accurate and in-line with the original paper. As the final note, late or so-called conceptual integration is perhaps of less interest for the community as it is often very specific to the research question. I would rather focus on late, model-based or statistical integration which might of higher value for most of the community.
Some other points:
Line 81: … for multi-omics analyses (e.g. Principal Components Analysis and Student’s t-test) could be strongly 82 affected by the presence of outliers [6,7]....
Frankly, your choice of method is sometimes odd. t-test is not a method of data integration and I can even argue about the PCA.
Line 96: … For example, scaling analyte measurements is more critical when applying a priori integration …
In fact, it is recommended to scale your data withing each omics but model with different descriptions across omics platform.
Line 98: For example, a priori integration requires the measurements to be collected in the same biospecimens (e.g. tissue, blood, etc.) or individuals to allow measurements to be matched to the same sample, while a posteriori integration does not.
I agree with you in most parts but this is not a general thing. Subspace and manifold alignment can be used without correspondence in “priori” multi-omics data analysis
Line 370: .. in the omics space, we direct the reader to [126]. Particularly relevant for multi-omics analyses, Karathanasis et al. [127] developed a method for combining results of hypothesis testing applied to separate omic modalities while accounting for correlations between the omic modalities…
This is the example of where i believe you should actually focus. Instead of writing about t-test, I would put the focus on the NPC method present by Karathanasis
Line 399: omic data. One common class of multivariate method is PLS-DA (Partial Least Squares Discriminant 400 Analysis). PLS-DA can be thought of as a semi-supervised variation of PCA
PLS and its variant (-DA) are certainly supervised methods. Please check the details.
Line 548: ... in the database. When multiple types of analytes are input into ORA analyses, the p-values resulting from analysis of each analyte independently are combined using Fisher’s method [194], which does not penalize an analyte type that has fewer annotations given a particular pathway. This approach is readily available in various software…
Yes, correct but you can always calculate weighted fusion using, for example, Stouffer's Z thus correcting the number of members of each pathways ...
Reviewer 4 Report
Strengths: Eicher et al., provide a detailed explanation of multi-omics analytical methods including pros and cons in order to provide a rubric for researcher to determine which method is most appropriate for a given analysis. This is a comprehensive description of the requirements and limitations of many of the most commonly used analytic methods in large data set publications. Given the prevalence of such data sets in modern research, the tendency for publications to favor "big data" graphic, and the importance such data has in revealing patterns and mechanisms in complicated biologic function, it is critical that researchers be able to critically interpret such data in publication, and in their own research.
Weaknesses: The organization of examples into tables is useful however the tables themselves specifically the "functionality" column are overly-wordy, and largely redundant with the organization of the main text. These tables could be more functional if the authors included pro/cons, or requirements/limitations as a column(s).
Round 2
Reviewer 3 Report
Dear Authors,
You have done a great job in improving the manuscript.
In my opinion, the manuscript is suitable for publication now.
Regarding your comment:
We thank the reviewer for this feedback. We note that, in the context of our manuscript, the term “a priori” refers to integration that directly follows preprocessing of the data, and not integration that depends on building single-omic models. While combining models of separate omics modalities on a manifold or subspace can be used as a method of integration, we do not include this in our definition of “a priori” techniques.
Please note that the vast majority of the model-based integration methods, including the ones you present in your manuscript (in fact almost all of them except the network-based integration), use subspace creation to do the integration. This is in fact "a prior" integration and the main technical difference is how this subspace is constructed.
-
I was perhaps now clear enough in my last point regarding the pathway analysis. Both Fisher’s and Stouffer’s method can be weighted to take into account the number of annotations you have in a particular pathway thus suffering less to the unbalanced annotations. It is just that mathematically, Stouffer’s method is easier to work with since you effectively derived the fused z from the normal distribution.
Good luck.